# Does Sperm *SNRPN* Methylation Change with Fertility Status and Age? A Systematic Review and Meta-Regression Analysis

**DOI:** 10.3390/biomedicines12020445

**Published:** 2024-02-16

**Authors:** Claudia Leanza, Rossella Cannarella, Federica Barbagallo, Carmelo Gusmano, Aldo E. Calogero

**Affiliations:** 1Department of Clinical and Experimental Medicine, University of Catania, 95123 Catania, Italy; claudia.leanza.95@gmail.com (C.L.); federica.barbagallo@phd.unict.it (F.B.); carmelo.gusmano@yahoo.it (C.G.); aldo.calogero@unict.it (A.E.C.); 2Glickman Urological and Kidney Institute, Cleveland Clinic Foundation, Cleveland, OH 44195, USA

**Keywords:** *Small Nuclear Ribonucleoprotein Polypeptide N* (*SNRPN*), DNA methylation, oligozoospermia, infertility, epigenetics

## Abstract

**Background:** The *Small Nuclear Ribonucleoprotein Polypeptide N* (*SNRPN*) gene is a paternally expressed imprinted gene, whose abnormal methylation appears to be associated with syndromes associated with the use of assisted reproductive techniques (ART), such as Angelman and Prader–Willi. Data present in the literature suggest the association between aberrant sperm *SNRPN* gene methylation and abnormal sperm parameters. The latest meta-analysis on the methylation pattern of this gene in spermatozoa of infertile patients published in 2017 reported a higher degree of methylation in the spermatozoa of infertile patients compared to fertile controls. **Objectives:** Here we provide an updated and comprehensive systematic review and meta-analysis of the sperm methylation pattern of the *SNRPN* gene in patients with abnormal sperm parameters/infertility compared to men with normal sperm parameters/fertile. For the first time in the literature, we performed a meta-regression analysis to evaluate whether age or sperm concentration could influence the methylation status of this gene at the sperm level. **Methods:** This meta-analysis was registered in PROSPERO (n. CRD42023397056). The Preferred Reporting Items for Systematic Reviews and Meta-Analysis Protocols (PRISMA-P) and the MOOSE guidelines for meta-analyses and systematic reviews of observational studies were strictly followed in our meta-analysis. According to our Population Exposure Comparison Outcome (PECO) question, we included data from original articles assessing the levels of *SNRPN* gene methylation at the sperm level in infertile patients or patients with abnormalities in one or more sperm parameters compared to fertile or normozoospermic men. **Results:** Only six of 354 screened studies were included in the quantitative synthesis. Our analysis showed significantly higher levels of *SNRPN* gene methylation in patients compared to controls. However, significant heterogeneity was found between studies. In sensitivity analysis, no studies were sensitive enough to skew the results. The Egger test showed no publication bias. In the meta-regression analysis, the results were independent of age and sperm concentration in the overall population. The same results were found in the control group. However, when analyzing the patient group, a direct correlation was found between *SNRPN* methylation and age, indicating that the degree of methylation of the *SNRPN* gene increases with advancing age. **Conclusions:** Fertility status or abnormality of sperm parameters is associated with a change in the methylation pattern of the *SNRPN* gene, with higher levels found in infertile patients or those with abnormal sperm parameters compared to fertile men or men with normal sperm parameters. In the group of infertile patients/patients with abnormal sperm parameters, age was directly correlated to the degree of *SNRPN* methylation, highlighting the presence of a mechanism that explains the age-related altered sperm quality and the risk of ART. Despite some limitations present in the analyzed studies, our results support the inclusion of *SNRPN* methylation in the genetic panel of prospective studies aimed at identifying the most representative and cost-effective genes to analyze in couples who want to undergo ART.

## 1. Introduction

Infertility is defined as the inability to achieve a pregnancy after 1–2 years of unprotected sexual intercourse and affects up to 15% of couples worldwide [1]. It can derive from male and/or female factors and the former contribute to couple infertility in approximately 50% of cases and are the sole cause in approximately 30% of couples [1]. Male infertility is usually diagnosed in the presence of alterations in sperm parameters (oligozoospermia, asthenozoospermia, teratozoospermia, or a combination of them), but based on these abnormalities it is not always possible to identify the cause, despite an accurate diagnostic process. In these cases, infertility is defined as “idiopathic” which can be present in up to 70% of cases [2].

Normal sperm parameters do not always coincide with fertility; in other words, infertility is not always associated with abnormal sperm parameters [3]. Therefore, researchers’ attention has turned to the genetic and epigenetic causes of male infertility. Epigenetics is defined as the combination of meiotic and mitotic molecular changes that regulate gene expression without changing the DNA sequence [4]. The most common epigenetic modifications occurring in spermatozoa are DNA methylation, histone modification, and chromatin remodeling [3,5,6]. DNA methylation is involved in mammalian spermatogenesis [7]. Indeed, when undergoing epigenetic changes, particularly hypermethylation, several genes are associated with abnormal sperm parameters or male infertility. These genes include: *Paired box 8, Methylenetetrahydrofolate reductase, Insulin-like growth factor 2, H19*, *Stratifin*, *Neurotrophin 3*, *Harvey rat sarcoma virus, Maternally expressed gene 3*, *Ras protein specific guanine nucleotide releasing factor 1*, *JmjC-domain-containing histone demethylase 2A*, *DIRAS family, GTP-binding RAS-like protein 3*, *Pleomorphic adenoma gene 1*, *Potassium voltage-gated channel subfamily Q member 1*, *Small Nuclear Ribonucleoprotein Polypeptide N, Long QT intronic transcript 1*, and *Mesoderm specific transcript* [5]. In addition to the above-mentioned evidence relating abnormal methylation of the *SNRPN* gene with altered sperm parameters or male infertility, the methylation status of the *SNRPN* gene has also been associated with the onset of imprinting disorders, such as Prader–Willi and Angelman syndromes.

Indeed, the interest in epigenetics lies in both its diagnostic and prognostic potential during the management of infertile couples. Consequently, not only can it explain cases of apparently idiopathic infertility or unexplained pregnancy loss, but it can also have a prognostic value regarding the outcome of assisted reproductive techniques (ART), also in terms of the health of the offspring [3,8]. Few data in the literature suggest the presence of DNA methylation abnormalities in the sperm of partners of women with recurrent pregnancy loss. In this regard, Khambata and colleagues conducted a case-control study involving 112 couples with a clinical history of recurrent pregnancy loss and found aberrations in the sperm DNA methylation of imprinted genes. These include insulin-like growth factor 2-H19 differentially methylated region (DMR), intergenic differentially methylated region, mesoderm specific transcript, zinc finger protein (which regulates apoptosis and cell cycle arrest), DMR in intron 10 of *Potassium Voltage-Gated Channel Subfamily Q Member 1* gene, paternally expressed gene 3, and paternally expressed gene 10 [9]. These findings highlight the importance of evaluating paternal genetic and epigenetic factors in the diagnostic process to identify the causes of idiopathic recurrent pregnancy loss. Other evidence, however, indicates that the DNA methylation pattern of offspring conceived through ART may be different from that of naturally conceived progeny. Indeed, Cannarella and colleagues more recently conducted a meta-analysis of 50 studies and found significantly reduced methylation of a CTCF-binding site in the *H19* gene, the CTCF3, in offspring conceived through ART compared to those conceived spontaneously [10]. Epigenetic damage in children conceived though ART could result from the manipulation of gametes during these techniques, considering that the timing of the methylation pattern of paternal and maternal alleles during embryogenesis coincides with that of in vitro fertilization (IVF), intracytoplasmic sperm injection (ICSI), embryo culture, and embryo transfer [10,11]. However, this is not the only possible explanation, since the transmission to the embryo of a compromised epigenetic pattern already existing in the paternal gamete cannot be excluded [12], especially in the case of imprinted genes. Indeed, genome-wide demethylation and de novo methylation take place in the preimplantation embryo, except in imprinted genes, whose methylation pattern is not altered to allow for parent-specific expression [13,14]. Therefore, an abnormal methylation pattern in imprinted genes is transferred to offspring.

The *SNRPN* gene (OMIM 182279) is a bicistronic imprinted gene located within an imprinted gene cluster on chromosome 15, associated with Prader–Willi syndrome (PWS) and Angelman syndrome (AS), two clinically different neurogenetic disorders caused by the lack of the maternal or paternal 15q11–q13 allele, respectively. *SNRPN* encodes two polypeptides, which are the SmN splicing factor, involved in RNA processing, and the SNRPN upstream reading frame (SNURF) polypeptide. The *SNRPN* gene is expressed exclusively from the paternally inherited chromosome and is expressed mainly in the heart and brain [15].

Scarce data are available on the methylation of the *SNRPN* gene at the sperm level and pregnancy outcome level [8,9], while more data are available on the methylation pattern in spermatozoa of patients with abnormal compared to normal conventional sperm parameters. The latest meta-analysis on this topic was published in 2017 and showed that methylation is significantly higher in the spermatozoa of patients with abnormal sperm parameters compared to normozoospermic men [16].

Currently, it is necessary to take into account the impact that paternal age can have on reproductive outcomes: men’s age at fatherhood has increased in industrialized countries for socioeconomic/cultural reasons, and the spread of ART and advanced paternal age appear to be associated with ART outcomes and offspring health outcomes [17,18,19]. Emerging evidence indicates a negative impact of age on sperm quality as demonstrated by its effect on several parameters such as semen volume, percentage motility, progressive motility, and normal morphology, but also sperm DNA fragmentation [20,21]. However, no comprehensive meta-regression analysis has been performed to understand whether the sperm methylation status of *SNRPN* changes as a function of age.

These premises lead us to hypothesize that the methylation status of the *SNRPN* gene at the sperm level may play a role in male infertility. Furthermore, we evaluated the possibility that the methylation pattern of this gene changes with advancing age and therefore could be among the factors responsible for the decline in sperm quality that occurs with aging. To achieve these goals, this systematic review and meta-analysis aim was undertaken to evaluate *SNRPN* gene methylation patterns in patients with abnormal versus normal sperm conventional parameters and, subsequently, assess whether age influences *SNRPN* methylation at the sperm level. To accomplish this, we conducted a meta-regression analysis (the first in the literature on this topic), to investigate the association between *SNRPN* gene methylation status, age, and sperm concentration in patients and controls.

## 2. Materials and Methods

### 2.1. Search Strategy

This meta-analysis was registered on PROSPERO (n. CRD42023397056). In conducting our meta-analysis, the Preferred Reporting Items for Systematic Reviews and Meta-Analysis Protocols (PRISMA-P) [22] and the MOOSE guidelines for Meta-Analyses and Systematic Reviews of Observational Studies [23] were rigorously followed.

The Scopus, PubMed, Cochrane, and Embase databases were searched up to December 2023. A combination of MeSH terms and keywords was used for the search strategy: “*SNRPN*”, “gene methylation”, “fertilization rate”, “sperm DNA fragmentation”, “assisted reproductive technique”, “pregnancy rate”, “abortion”, and “miscarriage”. Further searches were performed manually using the reference lists of relevant studies. No linguistic restrictions were applied in any literature search.

### 2.2. Selection Criteria

Eligibility of the studies was assessed using the PECOS (Population, Exposure, Comparison/Comparator, Outcome, Study type) model system [24]. Studies that included women, adolescents, or patients with azoospermia were excluded. Instead, original articles containing information on the methylation status of the *SNRPN* gene at the sperm level of patients with abnormal sperm parameters (oligo-, astheno-, and/or terato-zoospermia) or infertility compared to controls with normal sperm parameters (normozoospermia) or fertile were included (Table 1).

### 2.3. Data Extraction

The following data were extracted: study design, number of patients and controls, age, sperm concentration, fertility status, and *SNRPN* gene methylation. Data were extracted by one researcher (C.L.) and then verified by a second researcher (R.C.). A third, more senior researcher (A.E.C.) resolved any disagreements.

### 2.4. Quality Assessment

The Cambridge Quality Checklist [25] was used to assess the quality of evidence (QoE) of the included studies by C.L. It is a simple, fairly objective, and reproducible checklist, structured into three domains that include the quality of study correlates, risk factors, and a random risk factors assessment. The first evaluates the appropriateness of sampling methods and sample size, the quality of outcome, and measurement of correlates, and consists of five items. Each of them can be assigned a score of 0 or 1, for a total score of 5. The study design is categorized from the risk factors checklist into cross-sectional, retrospective, or prospective. These correspond to a score of 1, 2, or 3, respectively, with higher scores identifying those studies with appropriate time-ordered data. Finally, the casual risk factors checklist allows us to differentiate uncontrolled studies from controlled ones, also considering the presence of randomization. The score of this domain ranges from 1 (cross-sectional study without control group) to 7 (randomized clinical trial study). The total Cambridge Quality Checklist score is the sum of the scores from the three domains. The higher the score, the higher the quality of the study.

### 2.5. Statistical Analysis

Quantitative data analysis was performed using the Comprehensive Meta-Analysis Software (Version 3) (Biostat Inc., Englewood, NJ, USA) for meta-analysis of quantitative data. Heterogeneity between studies was assessed with Cochran’s Q test and heterogeneity index (I^2^), with a *p*-value < 0.05 considered statistically significant. The pooled effect size was calculated using fixed or random effects models in the absence or presence of inter-study heterogeneity, respectively. For sensitivity analysis, the pooled effect size and corresponding 95% confidence interval (CI) were calculated after excluding one study at a time. A study that resulted in the change of inference following its exclusion was labeled a ‘sensitive study’. Publication bias was analyzed qualitatively by the asymmetry of the funnel plot, which suggested some missing studies on one side of the plot. For the quantitative analysis, an Egger test was used, which evaluates the statistical significance of publication bias.

## 3. Results

A total of 354 articles were retrieved. After the exclusion of duplicates, 133 abstracts were assessed. Of these, 61 were found not pertinent after reading their abstracts, 10 were not original studies (7 were reviews, and 3 were a letter to the editor, a comment, or a book chapter), and 3 were animal studies. Therefore, 59 full-text articles underwent the assessment of eligibility, with the exclusion of 24 non-pertinent studies: 20 animal studies, 5 review articles, and 4 articles whose data could not be extracted. Finally, six studies satisfied the criteria for inclusion in the analysis (Figure 1; Table 2).

### 3.1. Quality of Evidence Results

The Cambridge Quality Checklist was used in the evaluation of the included studies. Out of a total score of 15, one scored 11, three scored 10, and two scored 9, indicating a high level of evidence for the majority of the included studies (Table 3).

### 3.2. Difference in SNRPN Methylation between Patients and Controls

Patients with abnormal sperm parameters/infertility showed significantly higher *SNRPN* methylation levels than fertile controls/men with normal sperm parameters (SMD 1.20, 95% CI: 0.47, 1.93, *p* < 0.001) (Figure 2). The random effect model was used since the presence of inter-study heterogeneity that was found at the Q-test (Q-value = 154.64; *p* < 0.001) and I^2^ = 94.83%. On sensitivity analysis, no study was sensitive enough to alter the results (Figure 3A). The analysis showed the absence of publication bias, as demonstrated by the Egger test (intercept 8.18, 95% CI: −0.81, 17.18, *p* = 0.07) and by the symmetry of the funnel plot (Figure 3B). These results support the robustness of the findings and the absence of sources of biases capable of modifying them.

### 3.3. Meta-Regression Analysis: Correlation between SNRPN Methylation and Age

To investigate the above-mentioned difference in *SNRPN* methylation status between cases and controls, we performed a meta-regression analysis correlating *SNRPN* gene methylation with mean age. In all populations, no significant correlation was found between *SNRPN* methylation and age (Figure 4A). A significant direct correlation between *SNRPN* methylation and age was found in the patient population (Figure 4B), indicating that the degree of methylation of this gene increases with age. On the other hand, no relationship was found in the control group (Figure 4C).

### 3.4. Meta-Regression Analysis: Correlation between SNRPN Methylation and Sperm Concentration

Considering the relationship between *SNRPN* methylation and sperm concentration, meta-regression analysis provided evidence for non-significant results in the overall population (Figure 4D), patients (Figure 4E), and controls (Figure 4F).

## 4. Discussion

The present systematic review and meta-analysis showed significantly higher methylation levels of the *SNRPN* gene in spermatozoa of patients with infertility/abnormal sperm parameters compared to fertile controls/normozoospermic men. Meta-regression analysis of the data showed the presence of a direct correlation between *SNRPN* methylation levels and age in the patient group.

The first finding supports what has already been reported in the previous meta-analysis on this topic [16], suggesting the association between increased *SNRPN* methylation status at sperm level and abnormal sperm parameters and/or male infertility. As far as we know, no data in the literature suggest the mechanism through which *SNRPN* hypermethylation impairs spermatogenesis or causes infertility.

The second finding, instead, might suggest *SNRPN* hypermethylation as a mechanism for the association between advanced age and impaired sperm quality [20,21]. Currently, advanced paternal age represents a significant problem, as the age at which a growing number of men approach fatherhood has increased for socioeconomic and cultural reasons in industrialized countries. The spread of ART may also have played a role in the increase in paternal age as it allows older couples to achieve pregnancies [32]. Numerous pieces of evidence in the literature suggest not only its correlation with abnormal conventional sperm parameters but also with an increased rate of sperm DNA fragmentation and miscarriage. For example, Johnson and colleagues conducted a meta-analysis including 90 studies, involving a total of 93,839 subjects, to evaluate the effect of male age on sperm parameters. The authors found a statistically significant decline associated with age in semen volume, percentage motility, progressive motility, normal morphology, and non-fragmented cells [20]. Furthermore, a recent systematic review and meta-analysis, including 19 prospective or retrospective studies, for a total of 40,668 subjects, aimed at investigating the impact of advanced paternal age and sperm DNA fragmentation, showed that older men have a higher rate of sperm DNA fragmentation index. The studies included in this systematic review were conducted on different categories of patients: 6 on fertile men and infertile patients, 4 on men with normozoospermia and subfertile patients, 4 only on infertile patients, 3 on men with normozoospermia, and 2 on a general population [32]. Several possible mechanisms have been suggested, such as structural changes of the male reproductive tract, including the narrowing of the seminiferous tubules, a decrease in function of reproductive accessory glands, and systemic disease related to aging, but also a decrease in the ability to repair cellular and tissue damage, increased oxidative stress, and inefficient apoptosis [18,33,34]. In particular, Singh and colleagues conducted a study on men aged between 20 to 57 years, aimed at investigating the association between age and DNA damage and sperm apoptosis. The authors reported an increase in sperm double-stranded DNA breaks with advancing age and also suggested an age-related decrease in sperm apoptosis, which may indicate a deterioration in the cell selection of healthy spermatozoa with age [34]. Advanced maternal age is a recognized factor in increasing the risk of miscarriage, but advanced paternal age may also play a role. Indeed, a meta-analysis including 9 studies showed an increased risk of miscarriage in the case of advanced paternal age after adjusting the data for maternal age [35]. More specifically, the most significant effect was found in the age category between 40 and 44 years [35]. Furthermore, several data in the literature suggest an association between age and ART outcome, pregnancy rate, and offspring health [18,19]. Regarding the relationship between advanced paternal age and ART outcomes, there is evidence of an increased risk of miscarriage after intrauterine insemination, IVF, and ICSI [18]. Regarding the association between advanced paternal age and offspring health, the most common adverse condition are musculoskeletal disorders, cleft palate, retinoblastoma, acute lymphoblastic leukemia, and neurodevelopmental disorders including autism and schizophrenia [19].

Some imprinted genes appear to be associated with ART outcome and pregnancy loss when aberrant methylation occurs. For example, male partners of couples with idiopathic recurrent pregnancy loss showed significantly lower H19 methylation levels and altered GLT2 methylation was associated with lower ART outcomes [8]. Evidence on the role of the methylation of the *SNRPN* gene in ART outcomes in humans is scarce.

More data are available on the role of *SNRPN* hypermethylation in spermatozoa and imprinted diseases in offspring, such as PWS. As mentioned above, the *SNRPN* gene is an imprinted gene located on chromosome 15 and is mainly expressed in the heart and brain [15]. In 1996, Glenn and colleagues conducted a study on solid tissue collected from multiple organs of patients with PWS and AS and showed functional imprinting of the human *SNRPN* gene through RT-PCR. They observed no expression of this gene in cultured skin fibroblast cells from PWS patients, whereas it was expressed in all AS patients and in normal controls. Furthermore, a parent-specific DNA methylation imprint in intron 5 of the *SNRPN* gene was demonstrated by the authors. This indicates the inheritance of parent-specific methylation of this gene [36]. Normally, the paternally derived *SNRPN* allele is unmethylated and acetylated, and therefore expressed, while the maternally derived is methylated and hypoacetylated, and therefore not expressed [37]. The absence of paternal *SNRPN* gene expression causes the PWS phenotype and this occurs mainly due to paternal deletions or maternal disomy of chromosome 15q11–q13 [38]. Notably, PWS patients lack *SNRPN* exon 1, which has been suggested to contain an imprint switch element from which the maternal and paternal epigenotypes of the 15q11–q13 domain originate [15]. Recent data suggest that the *SNURF-SNRPN* gene and two genes upstream of *SNURF-SNRPN* (*PWRN1* and *PWRN2*) are biallelically expressed in the testis and may play a role in imprinting, possibly keeping the paternal allele in an open chromatin conformation [39].

These findings suggest that an abnormal methylation of sperm *SNRPN*, in particular its high methylation and subsequent low expression, could cause imprinting disorders, mimicking the absence of the paternal allele in the offspring. Indeed, as mentioned before, the methylation pattern of imprinted genes has the characteristic of being maintained through genomic reprogramming that occurs during spermatogenesis and therefore of being transmitted to offspring, even if abnormal [12]. Male infertile patients or patients with abnormal sperm parameters are more likely to resort to ART, so the altered epigenetic state of the *SNRPN* gene (in this case a higher methylation and thus a lower expression) could be transmitted to offspring and lead to imprinted disorders such as PWS.

Similarly, older men, whose sperm *SNRPN* gene methylation we have shown to be higher, could transmit this epigenetic pattern onto their offspring, increasing the risk of offspring diseases. Evidence in the literature suggests an association between advanced paternal age and offspring mental disorders, including intellectual disability, autism, schizophrenia, and bipolar disorder [40]. In parallel, evidence suggests that chromosome segment 15q11–q13 is involved in autism spectrum disorders, and *SNRPN* as one of the autism-related genes [41]. Furthermore, evidence suggests that epigenetic dysregulation may have a role in the multifactorial etiology of autism spectrum disorder [41]. Based on this information, it would be important to understand whether the altered methylation status of *SNRPN* at the sperm level could represent a risk factor and a possible mechanism for autism spectrum disorders in the offspring of older men, as well as in those who have to resort to ART to overcome their infertility. If further studies confirm this association, the preliminary evaluation of *SNRPN* gene methylation status of spermatozoa will become indispensable during counseling for couples who must resort to ART, especially those with advanced paternal age.

The present study has some limitations, such as the low number of studies and the heterogeneity of data. In particular, the low number of studies did not allow us to sub-analyze the data based on the patient’s phenotype (i.e., oligozoospermia, asthenozoospermia, etc.). However, the methodology, such as the use of multiple databases to search for studies and the accuracy of data extraction, is the strength of this study.

## 5. Conclusions

Our results support the presence of higher methylation levels of the *SNRPN* gene in the spermatozoa of patients with infertility/abnormal sperm parameters and may suggest this epigenetic deterioration as a possible cause of otherwise unexplained male infertility. Through a meta-regression analysis, a direct association was found between age and higher sperm *SNRPN* methylation in infertile patients. Hypermethylation of *SNRPN* causes a lack of its expression. In turn, the absence of expression of the paternal allele is associated with PWS, an imprinting disorder. Considering the high prevalence of imprinted diseases among children conceived with ART [42], the role of the methylation status of the sperm *SNRPN* gene in the health outcomes of offspring conceived with ART should not be excluded. Furthermore, hypermethylation of sperm *SNRPN* occurring in older men might represent a possible mechanism of age-related impairment of sperm quality and provide insight into the pathogenesis of the risk of autism spectrum disorders in the offspring of fathers with advanced age.

## Figures and Tables

**Figure 1 biomedicines-12-00445-f001:**
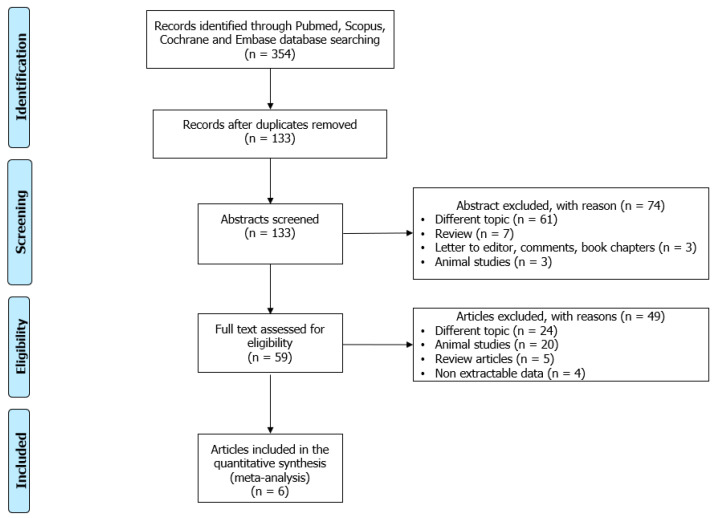
Preferred Reporting Items for Systematic Reviews and Meta-Analyses (PRISMA) flow-chart.

**Figure 2 biomedicines-12-00445-f002:**
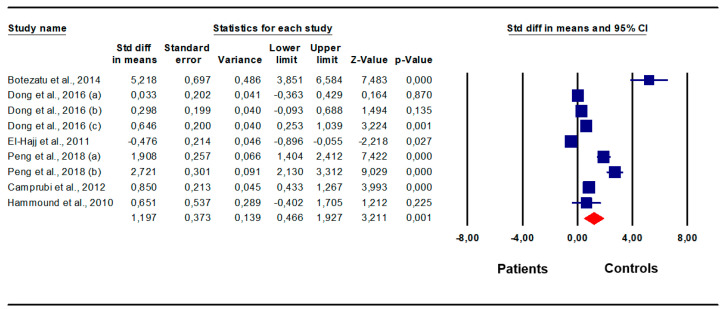
*SNRPN* methylation levels in patients vs. controls. Botezatu et al., 2014 [26]; Camprubi et al., 2012 [27]; Dong et al., 2016 [28]; El-Hajj et al., 2011 [29]; Hammound et al., 2010 [30], Peng et al., 2018 [31].

**Figure 3 biomedicines-12-00445-f003:**
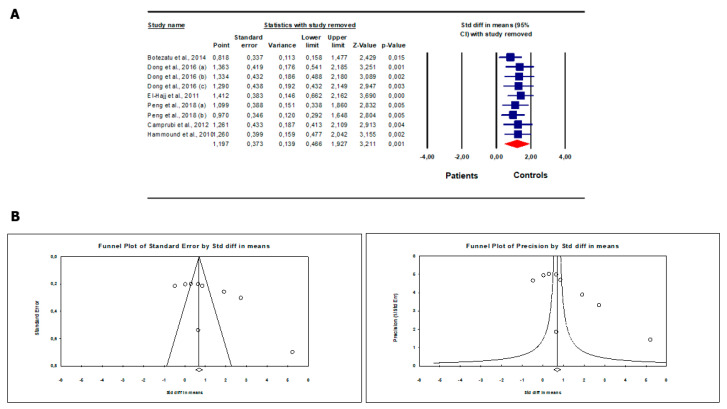
*SNRPN* gene methylation levels in infertile patients/patients with abnormal sperm parameters compared to fertile controls/men with normal sperm parameters: sensitivity analysis. (**A**) and analysis of publication bias (**B**). Botezatu et al., 2014 [26]; Camprubi et al., 2012 [27]; Dong et al., 2016 [28]; El-Hajj et al., 2011 [29]; Hammound et al., 2010 [30], Peng et al., 2018 [31].

**Figure 4 biomedicines-12-00445-f004:**
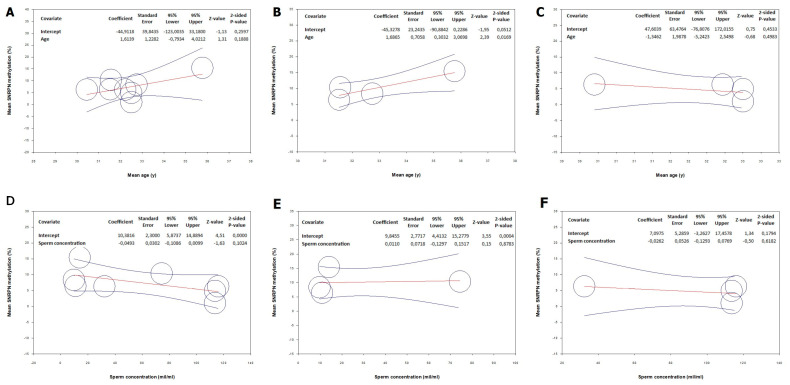
Meta-regression analysis between the standard difference in mean *SNRPN* gene methylation and the difference in mean age (**A**–**C**) and sperm concentration (**D**–**F**). The total population is included in (**A**,**D**). Only patients are included in (**B**,**E**). Only controls are included in (**C**,**F**).

**Table 1 biomedicines-12-00445-t001:** PECOS table reporting the inclusion and exclusion criteria [24].

	Inclusion Criteria	Exclusion Criteria
**Population**	Male patients	Women, male subjects younger than 18 years old, azoospermic patients
**Exposure**	Infertile or abnormal sperm parameters (oligo-, astheno-, and/or terato-zoospermia)	
**Comparison**	Fertile or normal sperm parameters (normozoospermia)	
**Outcomes**	*SNRPN* gene methylation	/
**Study type**	Randomized controlled studies, observational studies, case-control studies	In vitro studies, animal studies, review & meta-analyses, case reports, book chapters, editorials

Abbreviations. *SNRPN*, Small Nuclear Ribonucleoprotein Polypeptide N.

**Table 2 biomedicines-12-00445-t002:** Main characteristics of the studies included in the analysis.

Name	Type of Study	Cases			Controls		
		n	Characteristics	*SNRPN* Methylation Status	n	Characteristics	*SNRPN* Methylation Status
Botezatu et al., 2014 [26]	Prospective case-control study	27	Idiopathic infertile men	15.43 ± 2.62	11	N with F-factor infertility	1.76 ± 2.62
Camprubi et al., 2012 [27]	Prospective case-control study	15	N with infertility	1.2 ± 0.2	30	Fertile	1.03 ± 0.2
		1	O				
		8	A				
		30	T				
		1	OA				
		5	OT				
		31	AT				
		16	OAT				
Dong et al., 2016 [28]	Prospective case-control study	48	O	6.44 ± 3.72	50	NZ	6.32 ± 3.54
		52	A	7.74 ± 5.71			
		55	T	9.33 ± 5.48			
El-Hajj et al., 2011 [29]	Retrospective case-control study	106	M-factor infertility or combined M and F-factor infertility	3.8 ± 2.1	28	F-factor infertility	4.9 ± 3
Hammound et al., 2010 [30]	Retrospective case-control study	13	O	10 ± 9.9	5	Fertile	4.3 ± 3.5
Peng et al., 2018 [31]	Prospective case-control study	39	OA	8.36 ± 0.97	50	NZ	6.32 ± 1.14
		36	AT	10.37 ± 1.97			

Abbreviations. AT, Astheno-teratozoospermia; A, Asthenozoospermia; F: Female; M: Male; NZ,: Normozoospermia; O, oligozoospermia; OA, oligoasthenozoospermia; T, Teratozoospermia.

**Table 3 biomedicines-12-00445-t003:** Quality of evidence assessment of the included studies; results of the Cambridge Quality Checklist [25] (Murray et al., 2009).

Study Name	Type of Study	Cambridge Quality Checklists		
		Checklist for Correlates	Checklist for Risk Factors	Checklist for Causal Risk Factors
Botezatu et al., 2014 [26]	Prospective case-control study	2	3	5
Camprubi et al., 2012 [27]	Prospective case-control study	2	3	5
Dong et al., 2016 [28]	Prospective case-control study	3	3	5
El-Hajj et al., 2011 [29]	Retrospective case-control study	2	2	5
Hammound et al., 2010 [30]	Retrospective case-control study	2	2	5
Peng et al., 2018 [31]	Prospective case-control study	2	3	5

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
