# Peer review of "Does Sperm SNRPN Methylation Change with Fertility Status and Age? A Systematic Review and Meta-Regression Analysis"

_biomedicines, 2024, doi:10.3390/biomedicines12020445_

Round 1
Reviewer 1 Report
Comments and Suggestions for Authors
The Systematic Review and Meta-Regression Analysis performed by Leanza et al. is of value, and it can contribute to the evaluation of the epigenetic risk in assisted reproductive techniques (ART). They evaluated the SNRPN gene methylation patterns in patients with abnormal versus normal sperm conventional parameters and concluded that the presence of higher methylation levels of the SNRPN gene in spermatozoa of patients with infertility/abnormal sperm parameters suggests a possible cause of otherwise unexplained male infertility.
In my opinion, the paper can be accepted in its current form.
Author Response
Comment 1: The Systematic Review and Meta-Regression Analysis performed by Leanza et al. is of value, and it can contribute to the evaluation of the epigenetic risk in assisted reproductive techniques (ART). They evaluated the SNRPN gene methylation patterns in patients with abnormal versus normal sperm conventional parameters and concluded that the presence of higher methylation levels of the SNRPN gene in spermatozoa of patients with infertility/abnormal sperm parameters suggests a possible cause of otherwise unexplained male infertility. In my opinion, the paper can be accepted in its current form.
Answer to comment 1: Thank you very much for the time you took to review our article and for the words you spent for us.
Reviewer 2 Report
Comments and Suggestions for Authors
A very interesting manuscript on the topic of fertility status associated with a change in the methylation pattern of the SNRPN gene in semen. In my opinion a more interesting meta-analysis. The manuscript fully deserves to be published
Author Response
Comment 1: A very interesting manuscript on the topic of fertility status associated with a change in the methylation pattern of the SNRPN gene in semen. In my opinion a more interesting meta-analysis. The manuscript fully deserves to be published.
Answer to comment 1: We sincerely thank the reviewer for the time he took to read our article and for the words he wrote to us.
Reviewer 3 Report
Comments and Suggestions for Authors
The manuscript “Does sperm SNRPN methylation change with fertility status and age? A systematic review and meta-regression analysis” is devoted to giving information about the relationship between methylation and male infertility risk. I believe that the accumulation of information such as this manuscript is very important for future clinical applications. Although there are some limitations, I expect how much this method can contribute to infertility treatment. Especially, it needs additional schemes of their results to make it easier for the reader to read.
Some points have to be corrected.
Major points
1. It is necessary to explain why SNRPN was chosen among the many gene candidates.
2. After all, this paper organizes and summarizes the results reported until now. It needs to describe the limitations and problems of this manuscript.
3. The authors need to add the reliability of The Cambridge Quality Checklist.
4. Why did the authors use the Meta-Regression Analysis? What are the advantages and disadvantages of this method?
Minor points
1. Lines 53: Amend “of” to “in”.
Comments on the Quality of English LanguageMinor editing of English language required.
Author Response
Comment 1: The manuscript “Does sperm SNRPN methylation change with fertility status and age? A systematic review and meta-regression analysis” is devoted to giving information about the relationship between methylation and male infertility risk. I believe that the accumulation of information such as this manuscript is very important for future clinical applications. Although there are some limitations, I expect how much this method can contribute to infertility treatment. Especially, it needs additional schemes of their results to make it easier for the reader to read.
Answer to comment 1: We appreciate the reviewer’s time and effort in evaluating our article and her/his comments. We have double checked how the results were reported and think we were already written clearly, in a language common to other meta-analyses. We do not know what else to add “to make it easier for the reader to read”. If the editor thinks differently, please let us know what we can add to respond to this specific comment raised by the reviewer.
Comment 2: It is necessary to explain why SNRPN was chosen among the many gene candidates.
Answer to comment 2: Evidence in literature has already suggested a change in SNRPN methylation in sperm from infertile patients compared to fertile controls, so this represents a relevant target to focus on. Furthermore, it is an imprinted gene, thus impaired methylation at the sperm level can be transmitted to the offspring, with consequences for their health. Indeed, altered imprinting of SNRPN has been associated with Prader-Willi and Angelmann syndromes. This had already been explained in lines 94-114 of the previous version of the manuscript. However, considering the Reviewer’s comment, we have added lines 73-77 to further clarify this aspect.
Comment 3: After all, this paper organizes and summarizes the results reported until now. It needs to describe the limitations and problems of this manuscript.
Answer to comment 3: Done as requested. Please see lines 303-307 which read as follows: “The present study has some limitations, such as the low number of studies and the heterogeneity of data. In particular, the low number of studies did not allow us to sub-analyze the data based on the patient’s phenotype (i.e., oligozoospermia, asthenozoospermia, etc.). However, the methodology, such as the use of multiple databases to search for studies and the accuracy of data extraction are the strengths of this study”.
Comment 4: The authors need to add the reliability of The Cambridge Quality Checklist.
Answer to comment 4: The Cambridge Quality Checklist is a tool that allows us to evaluate the quality of scientific studies. It is divided into three sub-checklists to evaluate the study correlates, risk factors, and causal risk factors. We chose this checklist among others because of its objectivity in assessing of the three domains and also for its reproducibility since it is quite simple to use. Please see lines 153-165.
Comment 5: Why did the authors use the Meta-Regression Analysis? What are the advantages and disadvantages of this method?
Answer to comment 5: In the literature, there is evidence of the association between sperm SNRPN gene methylation status and abnormal sperm parameters and/or infertility, but there is no evidence of its association with age. Meta-regression analysis allows not only to correct an association for confounding factors but also to analyze the correlation between two variables. We hypothesized that age and sperm concentration could influence sperm SNRPN methylation status and performed a meta-regression analysis to investigate these associations. We found a direct correlation between SNRPN methylation levels and age in the patient group. In contrast, sperm concentration did not influence the associations mentioned above (please, see lines 121-123). This has never been investigated so far. We believe these findings are interesting because they could allow a better understanding of the factors responsible for the change in the methylation status of the SNRPN gene and possibly that of other genes.
Comment 6: Minor points:
- Lines 53: Amend “of” to “in”
- Minor editing of English language required.
Answer to comment 6: Done as requested. Thank you.
Round 2
Reviewer 3 Report
Comments and Suggestions for Authors
I think that the revised manuscript has been improved.